# Molecular mechanism and structure-guided humanization of a broadly neutralizing antibody against SFTSV

**Pinyi Yang**[1☉], **Xiaoli Wu**[2☉], **Hang Shang**[1☉], **Zixian Sun**[3], **Zhiying Wang**[2], **Zidan Song**[1], **Hong Yuan**[4], **Fei Deng**[2], **Shu Shen**[2]*, **Yu Guo**[1,3]*, **Nan Zhang**[1]*

**1** State Key Laboratory of Medicinal Chemical Biology and College of Life Sciences, Nankai University, Tianjin, China, **2** Key Laboratory of Virology and Biosafety and National Virus Resource Center, Wuhan Institute of Virology, Chinese Academy of Sciences, Wuhan, Hubei, China, **3** Guangzhou National Laboratory, Guangzhou, Guangdong, China, **4** Hangzhou Medimscience Biomedical Technology Co., Ltd., Hangzhou, Zhejiang, China

☉ These authors contributed equally to this work.
* shenshu@wh.iov.cn (SS); guoyu@nankai.edu.cn (YG); zhangnan@nankai.edu.cn (NZ)

**Data Availability Statement:** The structure coordinate of the SFTSV Gn head and mAb 40C10 complex has been deposited in the RCSB Protein Data Bank (PDB) with accession code PDB 8YXI.

## Abstract

Severe fever with thrombocytopenia syndrome virus (SFTSV) is a novel tick-borne bunyavirus that causes severe fever with thrombocytopenia syndrome (SFTS), with a high mortality rate of up to 30%. The envelope glycoproteins of SFTSV, glycoprotein N (Gn) and glycoprotein C (Gc), facilitate the recognition of host receptors and the process of membrane fusion, allowing the virus to enter host cells. We previously reported a monoclonal antibody, mAb 40C10, capable of neutralizing different genotypes of SFTSV and SFTSV-related viruses. However, the specific neutralization mechanism is poorly understood. In this study, we elucidated the high-resolution structure of the SFTSV Gn head domain in complex with mAb 40C10, confirming that the binding epitope in the domain I region of SFTSV Gn, and it represented that a novel binding epitope of SFTSV Gn was identified. Through in-depth structural and sequence analyses, we found that the binding sites of mAb 40C10 are relatively conserved among different genotypes of SFTSV and SFTSV-related Heartland virus and Guertu virus, elucidating the molecular mechanism underlying the broad-spectrum neutralizing activity of mAb 40C10. Furthermore, we humanized of mAb 40C10, which is originally of murine origin, to reduce its immunogenicity. The resulting nine humanized antibodies maintained potent affinity and neutralizing activity. One of the humanized antibodies exhibited neutralizing activity at picomolar $IC_{50}$ values and demonstrated effective therapeutic and protective effects in a mouse infection model. These findings provide a novel target for the future development of SFTSV vaccines or drugs and establish a foundation for the research and development of antibody therapeutics for clinical applications.

## Author summary

Severe fever with thrombocytopenia syndrome virus (SFTSV) infection has emerged as a global public health issue, and its incidence is increasing annually. Neutralizing antibodies

**Funding:** This work was supported by the Major Project of Guangzhou National Laboratory (SRPG22-002 to Y.G and SPRG22-003 to Z.S), the China Postdoctoral Science Foundation (2023M741824 to N.Z and GZC20232933 to X.W), the National Natural Science Foundation of China (32271256 to Y.G, U21A20180 to F.D and 32300146 to Z.S), the Foundation of State Key Laboratory of Component-based Chinese Medicine (CBCM2023104 to N.Z), and the Fundamental Research Funds for the Central Universities, Nankai University (030-63241622 to N.Z). The funders had no role in study design, data collection and analysis, decision to publish, or preparation of the manuscript.

**Competing interests:** FD, SS, XW, AM, YZ, ZW, TZ, SH and HWare the inventors in the pending patent applications filed on the reported antibody (mAb 40C10). HY is an employee of Hangzhou Medimscience Biomedical Technology Co., Ltd. Other authors declare no competing interests.

offer rapid and efficient therapeutic potential against viral infections. In this report, we elucidated the crystal structure of a highly effective and broad-spectrum murine neutralizing antibody, mAb 40C10, in complex with the SFTSV glycoprotein N (Gn) head domain. Through structural and sequence analyses, we revealed the molecular mechanism underlying the neutralization activity of mAb 40C10 and unveiled a new, potent, and broad-spectrum neutralizing epitope. Notably, this epitope is conserved across different genotypes of SFTSV and SFTSV-related viruses. The newly identified epitope provides valuable insights for reverse vaccine or drug design. Furthermore, we humanized mAb 40C10 and demonstrated the sustained efficacy of the humanized antibody in animal experiments, potentially advancing the development of clinical antibody therapeutics targeting SFTSV.

## Introduction

*Bunyavirales* is the largest and most diverse group of RNA viruses, consisting of 14 families and 63 genera, with over 552 members, according to the latest classification by the International Committee on Taxonomy of Viruses (ICTV) [1]. Bunyaviruses are widely distributed globally and can cause various epidemics that can have significant impacts on animal and human health and agriculture. Severe fever with thrombocytopenia syndrome virus (SFTSV) belongs to the genus *Bandavirus* in the family *Phenuiviridae*, order *Bunyavirales* (https://ictv. global/taxonomy/taxondetails?taxnode_id=202200166&taxon_name=Bandavirus% 20dabieense). It was first isolated in central China in 2009 and subsequently spread to multiple provinces in East China [2]. Clinical symptoms of SFTSV infection include fever, diarrhea, vomiting, thrombocytopenia, and leukopenia, with an initial high mortality rate of up to 30% [3,4]. The disease caused by this virus was officially named severe fever with thrombocytopenia syndrome (SFTS). Apart from China, cases of SFTS have been reported in Asian countries such as South Korea, Japan, and Vietnam, with several outbreaks occurring in Southeast Asia, where the mortality rate ranges from 6% to 27% [5–8]. The World Health Organization (WHO) prioritized SFTS as a disease of top priority for research [9].

SFTSV is a typical segmented negative-sense single-stranded RNA virus with a genome consisting of three circular RNA segments: small (S), medium (M), and large (L). These segments primarily encode nucleocapsid protein (NP), membrane precursor protein (GP), and RNA-dependent RNA polymerase (RdRp), respectively. Additionally, GP undergoes processing and modification by cellular proteases to yield two transmembrane glycoproteins, glycoprotein N (Gn) and glycoprotein C (Gc). Cryo-electron microscopy (cryo-EM) analysis of the structure of the SFTSV virion revealed that Gn and Gc form heterodimers that assemble into hexameric (hexon) and pentameric (penton) peplomers in a clustered manner on the virus membrane [10,11].

SFTSV is geographically distributed [12]. Phylogenetic analysis of the whole-genome sequences of SFTSV strains has revealed that they can be divided into Chinese and Japanese lineages [13]. The Chinese lineage predominantly includes C1–C5 genotypes, while the Japanese lineage encompasses three genotypes, J1–J3 [14,15]. The genetic diversity of SFTSV is notable, with studies suggesting that the segmented genome of the virus may promote genetic reassortment during virus replication within host cells, giving rise to new viral strains [5,13]. The genetic diversity and complexity of SFTSV present additional challenges for prevention and treatment efforts.

SFTSV infection is emerging as a pressing global public health issue, and its incidence is increasing annually. However, there are no vaccines, effective drugs, or specific treatment

methods available for SFTSV in clinical practice. Neutralizing antibodies can rapidly and efficiently treat viral infections. Currently, several neutralizing antibodies against SFTSV have been reported in various studies. These include the human antibodies MAb 4–5 [16,17] and Ab10 [18], as well as the nanobody SNB02 [19]. All three neutralizing antibodies target the surface glycoprotein Gn of SFTSV. The crystal structure of the SFTSV Gn head–MAb 4–5 complex has been resolved, elucidating the specific binding epitope and structural features of MAb 4–5, but the neutralizing activity of MAb 4–5 has only been validated *in vitro* [17]. Both Ab10 and SNB02 have demonstrated good neutralizing activity and protective effects both *in vivo* and *in vitro* [18,19]; however, the specific binding epitopes and structural details of neutralization remain unclear. Therefore, it is essential to screen for broad-spectrum neutralizing antibodies against various genotypes of SFTSV and characterize the resultant antibodies to establish a foundation for subsequent clinical drug development. In our previous study, we employed hybridoma technology to identify the murine monoclonal antibody 40C10 (mAb 40C10), which has strong neutralizing activity against multiple genotypes of SFTSV. *In vivo* experiments demonstrated the excellent delayed protection ability of the antibody [20].

Neutralizing antibodies can be acquired not only from diseased or recovered individuals but also through screening of mice immunized with virus particles or pathogenic proteins. Due to the disparity in the humoral immune response between mice and humans, it is possible for novel epitopes distinct from those originating in humans to emerge. However, the application of murine monoclonal antibodies in human clinical settings may lead to the recognition of foreign substances by the human body, resulting in a human anti-mouse antibody (HAMA) immune response [21]. In addition, murine monoclonal antibodies are unable to effectively trigger complement and Fc receptor effector functions, limiting their clinical application [22]. Therefore, the humanization of murine monoclonal antibodies is necessary. There are various methods for humanization, such as complementarity-determining region (CDR) grafting and resurfacing [23]. CDR grafting, involving the replacement of the mouse framework region (FR) with a human FR while preserving the mouse CDR and FR residues that may affect the activity of the mouse CDR, is commonly used and effective [24,25]. Key residues can be determined through the structure of antigen-antibody complexes. A previous study has reported that humanized Middle East respiratory syndrome coronavirus (MERS-CoV)-neutralizing antibody produced based on structural analysis have efficacy comparable to that of murine antibody [26]. Recently, after humanization, several murine monoclonal antibodies, such as Olokizumab, which targets Interleukin-6 (IL-6), Efalizumab, which targets lymphocyte function-associated antigen 1 (LFA-1), and Palivizumab, which targets the respiratory syncytial virus (RSV) F protein, have been marketed and utilized in clinical therapy [27,28]. This demonstrates the successful clinical application of humanized antibodies.

In this study, we determined the crystal structure of the SFTSV Gn head domain in complex with the fragment antigen-binding (Fab) portion of mAb 40C10 at a resolution of 2.4 Å. Based on the structure of the complex, we identified the binding epitope of mAb 40C10 on domain I of SFTSV Gn. We also found that this binding epitope is relatively conserved across various genotypes of SFTSV. This structural analysis provides insights into the broad-spectrum neutralization mechanism of mAb 40C10 and is the first to reveal a novel, potent, and broad-spectrum neutralizing epitope on SFTSV Gn. Furthermore, we performed mAb 40C10 humanization using CDR grafting. The humanized antibody was characterized, and the results demonstrated that the humanized antibody retained potent neutralization activity as well as animal protective and therapeutic efficacy in animals. These findings provide a basis for the clinical development of antibody drugs.

## Results

### Crystal structure of the SFTSV Gn head–mAb 40C10 complex

In the previous research, we screened and isolated the mAb 40C10, which has neutralizing activity at 2 ng/mL, using a hybridoma technique (Fig 1A). MAb 40C10 demonstrated the ability to neutralize different genotypes of SFTSV and exhibited a notable delayed protective effect *in vivo* [20]. Thus, mAb 40C10 displays a broad-spectrum and potent neutralizing effect. To further investigate the molecular mechanism underlying the neutralizing activity of mAb 40C10, we conducted structural biology studies.

After identifying the binding target of mAb 40C10 as the SFTSV Gn, we constructed the Gn head region, excluding the signal peptide (20–340), and expressed it using the baculovirus expression system. Additionally, we purified the mAb 40C10 IgG from mouse ascites, subjected it to enzymatic digestion using papain, and obtained the 40C10 Fab fragment. After incubating the Gn head with 40C10 Fab and purifying the complex, we ultimately obtained the stable Gn head–40C10 Fab complex and determined the crystal structure at 2.4 Å resolution (S1 Table).

In contrast to the majority of antigen-antibody interactions, mAb 40C10 primarily relies on the light chain to bind with Gn (Fig 1B and 1C), and the buried surface areas (BSAs) of the light and heavy chains are 495.4 Å$^2$ and 262.5 Å$^2$, respectively (S2 Table). The BSA of mAb 40C10 in contact with Gn is 757.9 Å$^2$, which is greater than that of the reported neutralizing antibody MAb 4–5 (614.8 Å$^2$). Three light-chain complementarity-determining regions (LCDRs), heavy-chain complementarity-determining region 3 (HCDR3) and light-chain framework region 2 (LFR2) together constitute the interaction network with Gn, which is mainly attributed to hydrogen bonds and hydrophobic interactions (Fig 1D). N31, Y50, S52, Y53, G66, Y67, N91, Y92, N93 and Y96 of the light chain, along with Y101, Y103, and R106 of the heavy chain form hydrogen bonds to interact with Gn. R106 of HCDR3 and D32 of LCDR1 form salt bridges with D102 and K111 of Gn, respectively (S3 Table). In addition, interactions are facilitated by a tight hydrophobic interaction network involving three LCDRs, HCDR3 and LFR2 with R62, H64, S65, Q66, V108, V109, K110, K113, G114, K147 and C156 of Gn. Notably, R106 of HCDR3 forms hydrogen bonds with glycan molecules at the N-linked glycosylation site N63 of Gn (Fig 1D).

### Analysis of the binding epitope of mAb 40C10 on SFTSV Gn

The Gn head region mainly contains a signal peptide and three domains, with domain II adjacent to the stem region. Through structural analysis, we identified that the binding epitope of mAb 40C10 is located on domain I of Gn, in contrast with the two neutralizing antibodies previously reported (Fig 2A). The binding epitope of MAb 4–5 is located in domain III of Gn [17], whereas Ab10 binds within the stem region and domain II adjacent to the stem region of Gn [18] (Fig 2A). Thus, this finding indicates the identification of a novel antigen binding site. Subsequently, we conducted a detailed analysis of residues on the interaction interface between mAb 40C10 and Gn. By measuring the affinity between mAb 40C10 and different Gn single-point mutants, we observed that the Gn mutants H64A, K111A, and K113A exhibited a 3–-5-fold decrease in affinity for mAb 40C10 compared that of the wild-type (WT) (S1 Fig), indicating the critical role of these residues in maintaining the interaction with mAb 40C10. In particular, the K111A mutant showed the greatest reduction in affinity, possibly due to its interaction with both the heavy and light chains (Fig 1C and 1D). SFTSV displays genotype diversity [5], featuring genotypes spanning C1–C5 and J1–J3 for the M segment that encodes the SFTSV Gn protein [13]. We conducted sequence alignments for strains of various

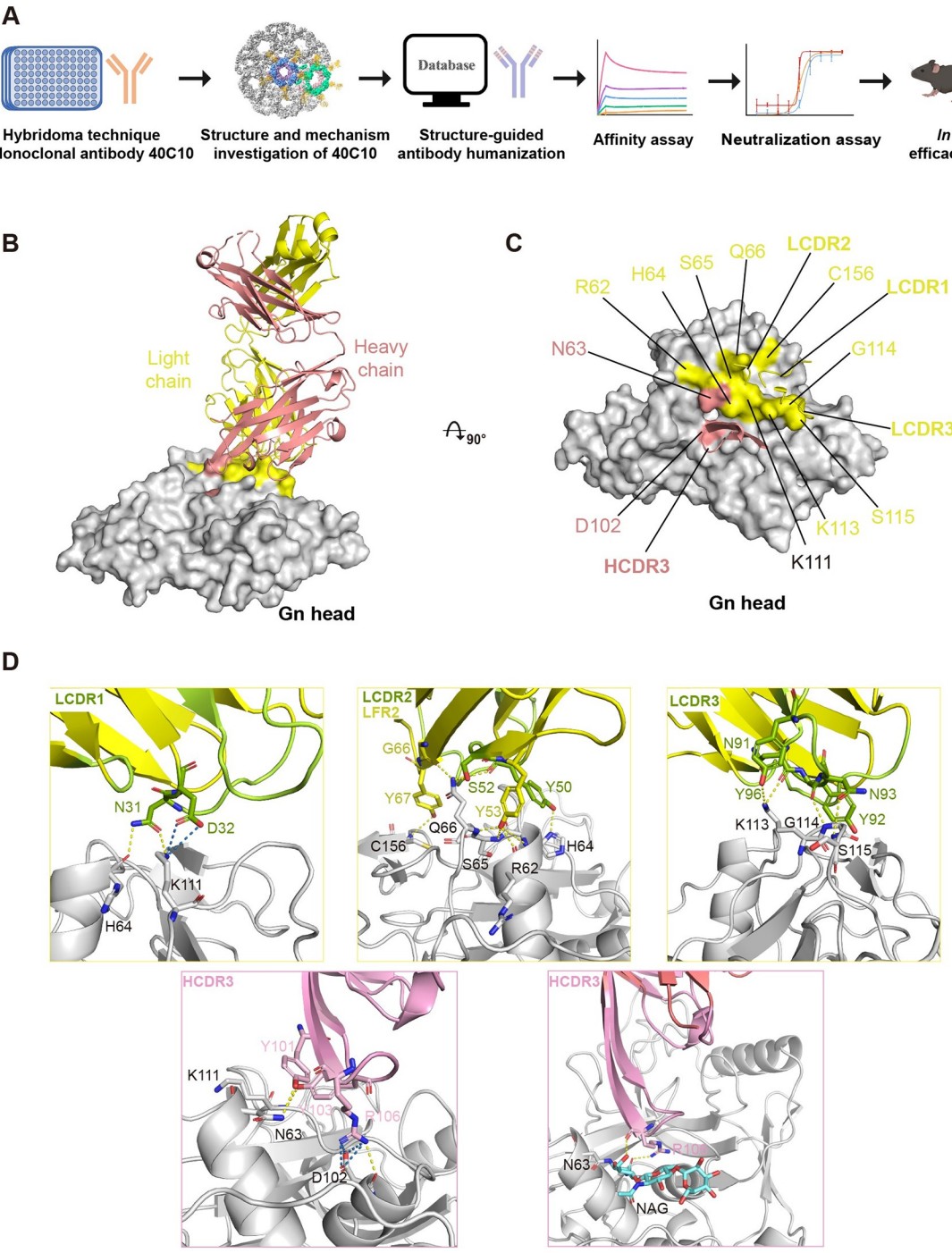

**Fig 1. Structural analysis of the SFTSV Gn head–mAb 40C10 complex.** (A) Schematic diagram showing the experimental design. (B) The overall structure of the SFTSV Gn head–mAb 40C10 complex. The mAb 40C10 heavy chain and light chain are colored salmon and yellow, respectively. The SFTSV Gn head is colored gray and displayed in surface representation. (C) The binding region of mAb 40C10 on the SFTSV Gn head. The region binding to the heavy chain of mAb 40C10 on the SFTSV Gn head is colored salmon, and the region binding to the light chain of mAb 40C10 on the SFTSV Gn head is colored yellow. The CDR loops of mAb 40C10 are colored as above. (D) The detailed interactions between the SFTSV Gn head and the LCDRs, HCDR3 and LFR2. The residues are shown as sticks and colored limon (LCDRs), yellow (LFR2) and pink (HCDR3). The glycosylation site on the SFTSV Gn head and the sugar molecules are presented in stick form and are colored gray and cyan, respectively. The yellow dashed lines represent hydrogen bonds, and the blue dashed lines represent salt bridges.

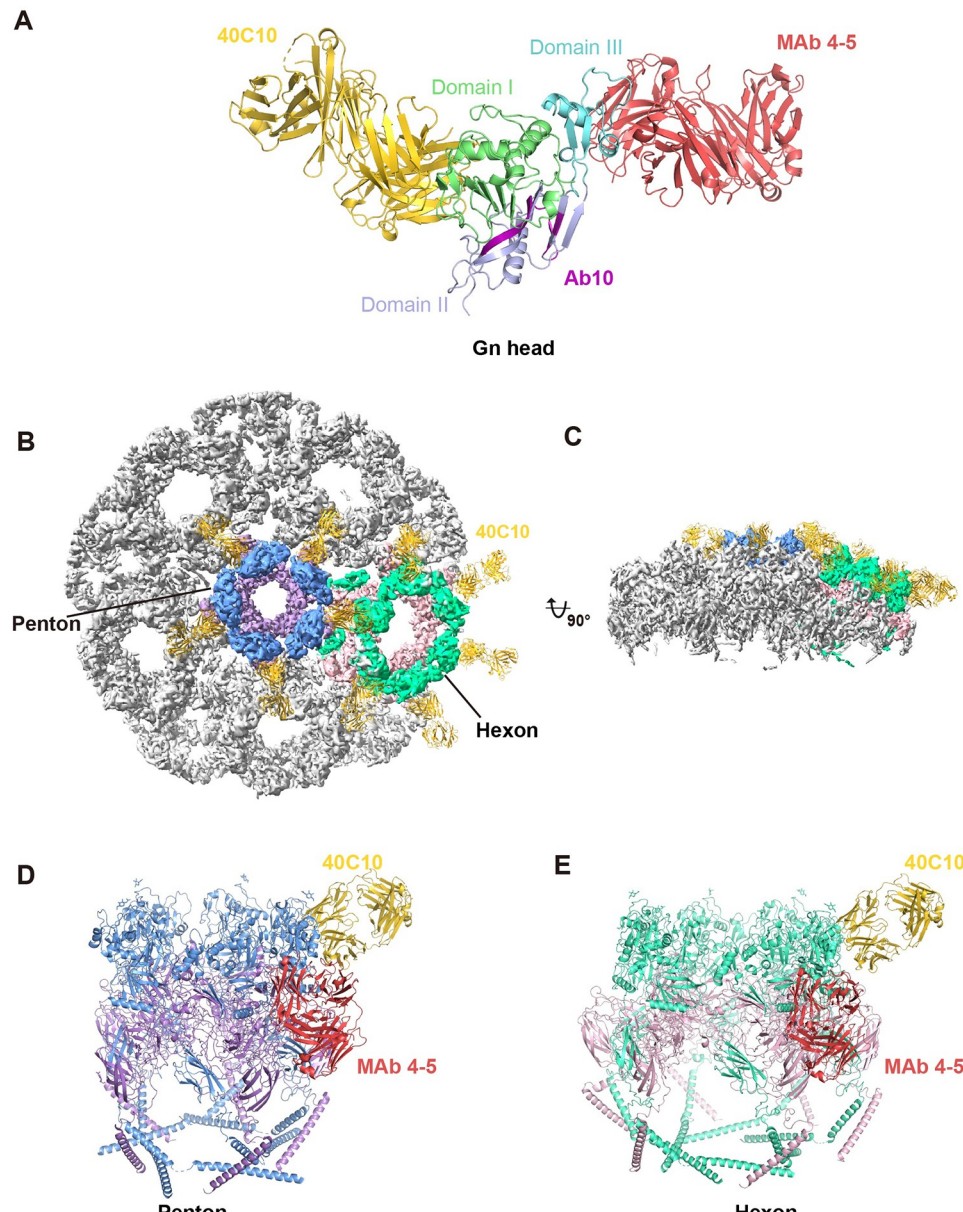

**Fig 2. Analysis of the binding epitope of mAb 40C10 on the SFTSV Gn head.** (A) The binding epitopes of mAb 40C10, MAb 4–5, and Ab10 on the SFTSV Gn head are depicted. The three domains of the SFTSV Gn head are colored pale green, light blue and aquamarine. MAb 40C10 is colored gold, MAb 4–5 is colored red, and the binding sites of Ab10 on the SFTSV Gn head are colored purple. (B and C) MAb 40C10 bound to the surface of the SFTSV virion (5-fold local reconstruction of SFTSV virion, EMD-35178), with penton and hexon highlighted in color. The Gn and Gc of penton are colored in cornflower blue and medium purple, respectively, and the Gn and Gc of hexon are colored medium spring green and light pink, respectively. (D and E) MAb 40C10 and MAb 4–5 bind to either the hexon or penton Gn protein. Penton, hexon, mAb 40C10 and MAb 4–5 are colored as above (PDB: 8I4T, 7X6W, and 7X72).

genotypes. Remarkably, the essential residues mediating binding to mAb 40C10 were conserved across the different genotypes (S2A, S2B and S3 Figs). Additionally, we identified the binding epitope in two newly discovered strains based on their genetic evolutionary relationships, Heartland virus (HRTV) and Guertu virus (GTV) [29,30], and found that the key

residues were also relatively conserved in these strains (S2C Fig). Furthermore, we consistently detected neutralizing activity of mAb 40C10 against different genotypes of SFTSV *in vitro*, and demonstrated cross-neutralization against both HRTV and GTV [20]. The above analysis elucidated the molecular mechanism by which mAb 40C10 exhibits broad-spectrum neutralization against various strains, from a structural perspective.

The SFTSV glycoproteins exist in different forms on the virion surface, with Gn-Gc heterodimers capable of assembling into hexons and pentons. To observe the specific binding of mAb 40C10 on the virion surface, we attempted to fit the structure of mAb 40C10 onto the hexon and penton of SFTSV based on the reported structure of the SFTSV virion [10,11]. The root mean square deviation (RMSD) values for the alignment of the Gn in the complex crystal structure with the Gn in the cryo-EM structure of the SFTSV virion were consistently less than 2 Å (S4 Fig) [10], indicating a high degree of structural similarity. This suggests that the fitting results are reliable. From the fitted structures, it was evident that mAb 40C10 binds well to the Gn protein in both the hexon and penton without any hindrance from the surroundings (Fig 2B and 2C). This is in contrast to MAb 4–5, which is influenced by steric hindrance from the Gn protein around the peplomers (Fig 2D and 2E). These observations may provide structural evidence for the potent neutralizing activity of mAb 40C10, which can reach picomolar concentrations.

## Humanization of mAb 40C10 by CDR grafting

MAb 40C10 was isolated from mice. To enhance its potential for clinical applications, we subjected mAb 40C10 to humanization (Fig 3A). Human germline counterparts with high sequence identity to both the heavy and light chains of mAb 40C10 were searched using NCBI IgBLAST. We selected the top three human germline V regions with the highest identity to the mAb 40C10 heavy chain variable region (VH): IGHV1-46*01 (66.0%), IGHV1-3*01 (66.0%), and IGHV1-2*06 (64.9%). For the mAb 40C10 light chain variable region (VL), we chose the top two germline V regions with the highest identity, IGKV1-33*01 (66.3%) and IGKV4-1*01 (66.3%). For the J region, IGHJ4*02 and IGKJ2*01 were selected, which differed by two and one residues from the VH and VL sequences of mAb 40C10, respectively (S5 Fig). Then, we utilized the FR of the human germline noted above as a template, and three CDRs from the mAb 40C10 heavy and light chains were grafted, resulting in three CDR-grafted heavy chains and two CDR-grafted light chains.

Based on the structure of the SFTSV Gn head–mAb 40C10 complex, we identified critical residues in the FR of templates that could impact the structure of the CDRs, resulting in a loss of affinity, and back mutations were applied to these residues in the CDR-grafted heavy chains and CDR-grafted light chains. Ultimately, we designed three heavy chains and three light chains. Compared to the sequence of mAb 40C10, the three heavy chains had 17, 23, and 25 mutation sites, while the three light chains had 15, 18, and 19 mutation sites. These designed chains were scored for their degree of humanization using the "Hu-mAb" website [31]. All the scores exceeded the threshold, indicating that the engineered chains achieved a high level of humanization (Figs 3B and S6). Finally, we paired the engineered heavy and light chains in all possible combinations, generating a total of 9 humanized antibodies, designated HAb-11, HAb-12, HAb-13, HAb-21, HAb-22, HAb-23, HAb-31, HAb-32, and HAb-33. In the humanized antibody designation, the first digit represents the heavy chain number, while the second digit represents the light chain number.

## Characterization of the humanized antibodies

To assess the affinity and neutralizing activity of the humanized antibodies, 9 humanized antibodies in human IgG1 form were expressed in HEK293 cells and purified using Protein A

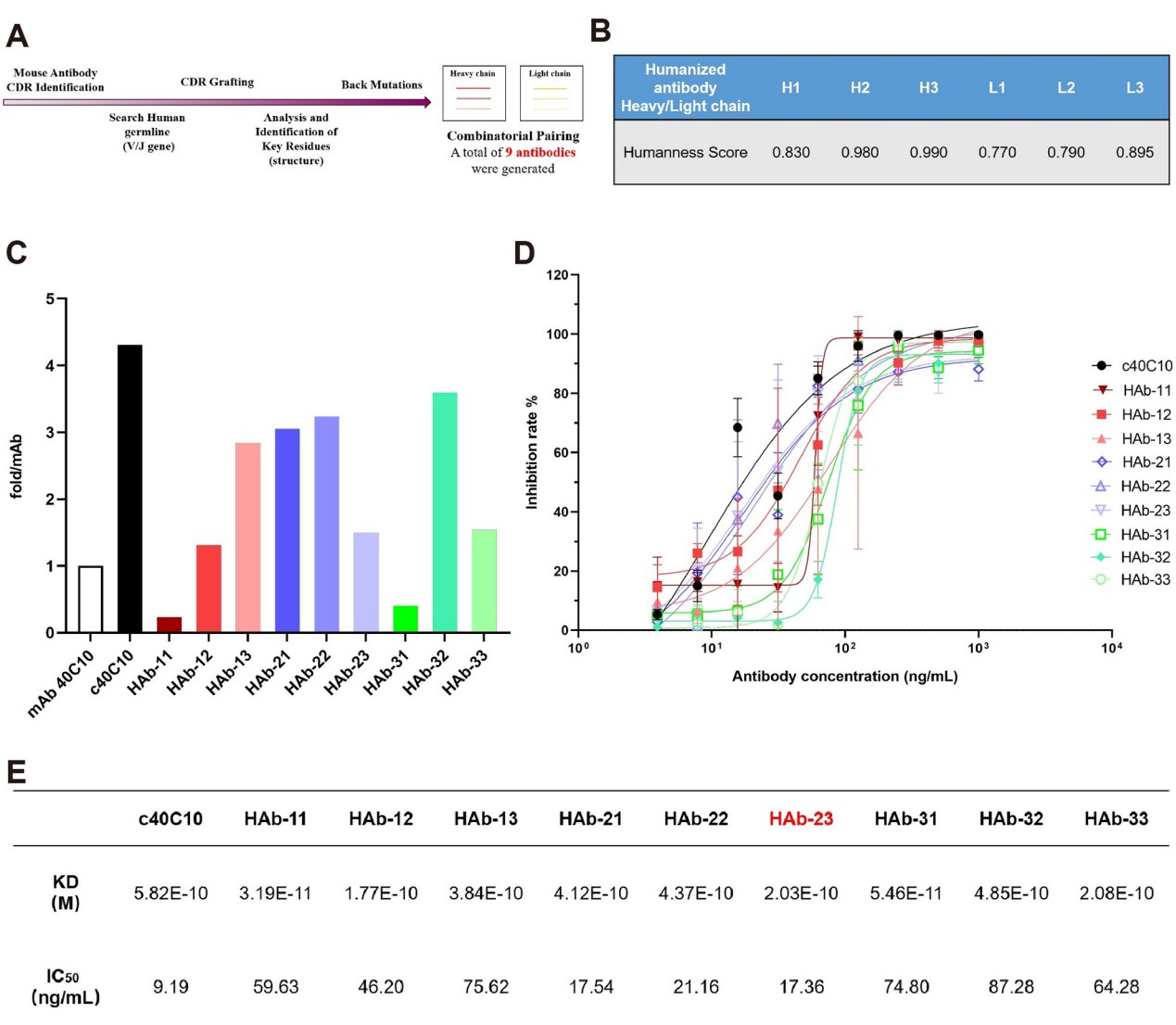

**Fig 3. Evaluation of humanized antibodies.** (A) Flowchart of antibody humanization. (B) Humanness score of the heavy and light chains of the humanized antibodies. (C) The ratio of the KD value of the humanized antibodies to the KD value of mAb 40C10. (D) The determination of neutralizing activity of the humanized antibodies. (E) The KD values and IC$_{50}$ values of the humanized antibodies.

affinity chromatography. For use as a control, we also expressed and purified a humanized IgG1 antibody with variable regions of the heavy and light chains from mAb 40C10, named c40C10. Then, we tested the binding affinity between the humanized antibodies and the SFTSV Gn head domain. According to surface plasmon resonance (SPR) analyses, the equilibrium dissociation constant (KD) for mAb 40C10 was $1.35 \times 10^{-10}$ M (S7 Fig). Using the KD of mAb 40C10 as a reference, the KD of c40C10 and the humanized antibodies differed by only approximately 4-fold at most, with the KD of some humanized antibodies (HAb-11 and HAb-31) being lower KD than that of mAb 40C10 (Figs 3C, 3E and S7). The binding affinity results showed that the humanized antibodies retained high binding activity to the SFTSV Gn head domain. We further evaluated the neutralization activity of the 9 humanized antibodies against SFTSV. Although there was a reduction in neutralization activity compared to that of mAb

40C10 and c40C10 [20], with differences ranging from several-fold to more than tenfold, the neutralizing activity remained in the picomolar range (Fig 3D and 3E). This result suggested that the humanized antibodies maintained significant neutralization activity.

Next, we selected the humanized antibody HAb-23, which had the best neutralizing activity, for *in vivo* efficacy evaluation.

### *In vivo* efficacy evaluation for the humanized antibody

To facilitate clinical application, the *in vivo* efficacy of the humanized antibody HAb-23 against SFTSV infection was evaluated in IFNAR[-/-] C57BL/6 mice. In this study, given that approximately half of the isolated SFTSV strains were of the C2 genotype, the HBGS13 strain (C2) was used to infect mice at the median lethal dose ($LD_{50}$) of 50. Prophylactic and therapeutic treatment groups were established and monitored for 12 days. (Fig 4A).

All mice in the prophylactic and therapeutic treament groups administered HAb-23 or mAb 40C10 survived, and their body weights ranged from 92.98% to 109.97% of the baseline (Fig 4B). In contrast, the mice in the PBS control group showed a decrease in body weight at 2 days post-infection (d.p.i.) and began to die at 5 d.p.i. (Fig 4B). Furthermore, the levels of viral RNA in the livers and spleens of mice from the HAb-23/mAb 40C10 prophylactic and therapeutic treatment groups were reduced at 5 d.p.i., as determined by quantitative real-time PCR (qRT–PCR) (Fig 4C).

In the PBS control group, hematoxylin-eosin (H&E) staining revealed inflammatory infiltration, scattered hemorrhage, hepatocellular necrosis in the liver, atrophy or loss of white pulp in the spleen, renal interstitial bleeding in the kidney, and thickening of alveolar septa in the lung (Fig 4D). Additionally, immunohistochemical examinations revealed the presence of SFTSV NP antigens in the liver and spleen. In contrast, for the HAb-23 prophylactic and therapeutic treatment groups, there was minimal inflammatory infiltration in the liver, slight loss of white pulp in the spleen, minimal interstitial bleeding in the kidney, and localized thickening of the alveolar septa in the lung at 5 d.p.i. (Fig 4D). Furthermore, all pathological symptoms were abrogated, and the organs had returned to a normal state by 12 d.p.i. (Fig 4D). In addition, only a small amount of SFTSV NP antigen was detected in the liver and spleen at 5 d.p.i., and SFTSV NP antigen was almost undetectable at 12 d.p.i. These results are similar to those observed for the mouse antibody mAb 40C10 group (S8 Fig). These findings suggest that the humanized antibody can still effectively protect against SFTSV infection *in vivo* when administered either prophylactically or therapeutically.

### Discussion

SFTS is an infectious disease caused by an emerging tick-borne virus known as SFTSV. SFTSV infection primarily results in symptoms such as fever, fatigue, thrombocytopenia, and leukopenia. In severe cases, it can lead to multiorgan failure and, ultimately, death, with a mortality rate as high as 30% [3]. SFTSV infection has become a globally significant public health concern, and several research groups have reported the development of DNA vaccines, mRNA vaccines, nanoparticle vaccines, neutralizing antibodies, and nanobodies targeting SFTSV [16,18,19,32–35]. However, there are currently no approved specific vaccines or antibody drugs against SFTSV for clinical use. We previously identified the mAb 40C10, which has potent neutralizing activity, and demonstrated its effective protection *in vivo* [20]. Nevertheless, the precise molecular mechanism of action for this antibody remains unclear. In this study, we analyzed the structure of the SFTSV Gn head–mAb 40C10 complex to identify the binding epitope of mAb 40C10. Remarkably, mAb 40C10 binds to domain I of SFTSV Gn, which is distinct from the previously identified binding site. This discovery revealed a novel

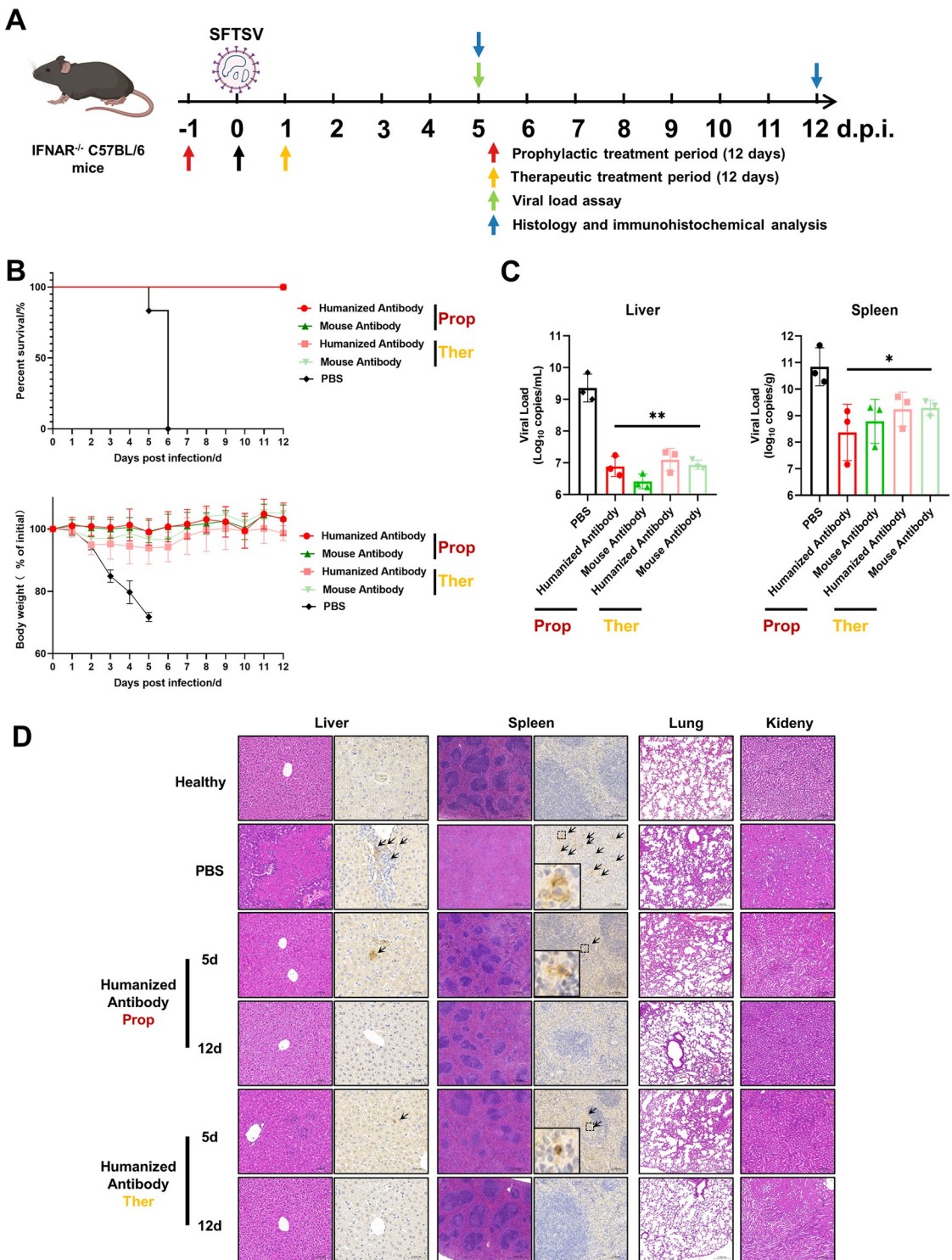

**Fig 4. Evaluation of the prophylactic and therapeutic efficacy of the humanized antibody *in vivo*.** (A) Experimental design for determining the prophylactic and therapeutic efficacy of the humanized antibody in IFNAR$^{-/-}$ C57BL/6 mice. (B) Survival rate (top) and body weight changes (bottom) of infected mice. (C) The viral loads in the livers and spleens of infected mice were measured by qRT–PCR at 5 d.p.i. (D) Histopathological analyses of humanized antibody HAb-23 treated or untreated infected mice.

binding epitope on SFTSV Gn. Besides, hydrogen bond interaction between mAb 40C10 and the glycan molecules of the N-linked glycosylation site N63 of Gn was observed. This glycosylation site is conserved across different genotypes of SFTSV (S2A and S2B Fig), and glycan modification at this site has been observed in several resolved structure of SFTSV Gn [11,17]. These observations suggest that the interaction between the antibody and SFTSV Gn is conserved and that glycosylation at this site may play a critical role in antibody binding.

Sequence alignment of different SFTSV genotypes shows that the binding epitope of mAb 40C10 is conserved, which is consistent with its effective neutralizing effect against various SFTSV genotypes *in vitro* [20]. Similarly, the critical binding epitopes of mAb 4–5 and Ab10 in the Gn head domain are conserved across different SFTSV genotypes (S9A and S9B Fig) [17,18], suggesting that mAb 4–5 and Ab10 might also broadly neutralize different SFTSV genotypes. Experimental data indicate that mAb 40C10 can cross-neutralize HRTV and GTV, with sequence analysis revealing that its binding epitope is relatively conserved in these SFTSV-related viruses. However, the critical binding epitopes of mAb 4–5 show lower conservation in HRTV and GTV (S9C Fig), indicating potentially weaker neutralization activity against these viruses. In contrast, the binding epitopes of Ab10 in the Gn head domain is relatively conserved in HRTV and GTV (S9C Fig), suggesting possible neutralization of these viruses. Further experimental validation is needed to confirm whether mAb 4–5 and Ab10 possess broad-spectrum neutralizing activity against different genotypes of SFTSV and SFTSV-related viruses. Based on the cryo-EM structure of the SFTSV virion [10,11], the structure and assembly of the glycoproteins Gn and Gc on the virus membrane can be clearly observed. In light of the characteristics of this structure, we investigated the binding regions of mAb 40C10, MAb 4–5 and Ab10 on SFTSV Gn. As shown in Fig 2A, MAb 4–5 binds to domain III of SFTSV Gn [17], and this interaction is influenced by steric hindrance from adjacent Gn/Gc heterodimers. Similarly, the binding epitope of Ab10 is located in domain II of SFTSV Gn, near the stem region [18], and this interaction is potentially affected by steric hindrance from the viral membrane. Therefore, we speculate that MAb 4–5 and Ab10 bind to SFTSV Gn during the membrane fusion process, specifically when Gn and Gc undergo conformational changes under acidic pH conditions in the endosome. In contrast, the binding site of mAb 40C10 is not affected by surrounding Gn/Gc heterodimers or the membrane, and each Gn in the hexon or penton can bind to mAb 40C10 (Fig 2B and 2C). Consequently, we hypothesize that mAb 40C10 may bind to Gn without Gn/Gc heterodimers undergoing conformational changes, thereby facilitating inhibition of viral infection.

Neutralizing antibodies against enveloped virus can exert their effects through multiple mechanisms [36]. Studies have shown that by analyzing the cryo-EM structures of three anti-Eastern equine encephalitis virus (EEEV) neutralizing human mAbs bound to EEEV, the distance between Fab regions bound to glycoprotein spikes can elucidate the mechanism of neutralization through intravirion or intervirion cross-links [37]. Similarly, mAb 40C10 may neutralize SFTSV, an icosahedral virus, through a mechanism involving virion cross-linking. We fitted mAb 40C10 to the surface of SFTSV at the 2-fold, 3-fold, and 5-fold symmetry axes and measured the distances between the CH1 domain ends of two adjacent mAb 40C10 Fabs. The distances ranged from 62.7 Å to 140.1 Å, all of which are greater than the distance between the two Fabs of an IgG molecule [38] (S10A, S10B and S10C Fig). This observation suggests that mAb 40C10 might form intervirion cross-links. To further substantiate this hypothesis, we measured the $IC_{50}$ values of mAb 40C10 in both its IgG and Fab forms (S10D Fig). The significantly better neutralization effect observed with the IgG form compared to the Fab form supports the hypothesis that the potential neutralization mechanism of mAb 40C10 involves intervirion cross-linking. Furthermore, we also fitted mAb 4–5 onto the SFTSV surface and found that its epitope is located internally rather than on the surface (S11 Fig). Thus, mAb 4–5

may not bind to Gn or form virion cross-links before the conformational changes in SFTSV glycoproteins.

To further understand the neutralizing mechanism of mAb 40C10, we first explored whether mAb 40C10 affects the virus attachment to cells. Experimental results showed that the amount of virus adsorbed to the cells in the PBS group and the mAb 40C10 group was similar, with no significant difference (S12A Fig). Previous studies have demonstrated that SFTSV enters cells through endocytosis into the cytoplasm, followed by membrane fusion with the late endosome membrane in the acidic environment, which releases the virus to complete the invasion process [39]. Based on these characteristics of SFTSV invasion of cells, we investigated whether the antibody affects virus endocytosis. During the early stage of virus adsorption, consistent with the previous steps, we added $NH_4Cl$ to the cell culture medium, which can inhibit the fusion of the virus with the endosome membrane [40]. The experimental results showed that the amount of virus entering the cells in the mAb 40C10 group was significantly lower than that in the PBS group (S12B Fig), suggesting that the antibody affected the internalization process of SFTSV. These observations, combined with the data suggesting that mAb 40C10 might form intervirion cross-links, lead us to hypothesize that mAb 40C10 may not interfere with the binding of the virus to cell receptors. Instead, it may form cross-links between virions outside the cell, causing them to aggregate. This aggregation could result in larger particle sizes or other effects that hinder viral internalization, thereby preventing the virus from entering and infecting the cells. However, this hypothesis requires further experimental validation.

Neutralizing antibodies directly target viral proteins, thereby affecting the interaction between the virus and host cells, ultimately inhibiting viral infection and preventing or treating viral infectious disease. The extensive development and clinical application of neutralizing antibodies during the coronavirus disease 2019 (COVID-19) pandemic have propelled their application in the treatment of viral diseases [41]. The screening of neutralizing antibodies from immunized mice is a common method. The application of murine monoclonal antibodies in clinical settings requires humanization to mitigate the inherent immunogenicity associated with murine antibodies. Several humanized antibodies have already been approved for clinical use [27,28]. Consequently, the humanization of antibodies has emerged as a crucial technological pathway in the development of clinical antibody therapeutics. In this study, we employed CDR grafting and backmutation to humanize the mouse antibody mAb 40C10. Nine humanized antibodies were designed, and they maintained favorable affinity and neutralizing activity. Furthermore, we selected the humanized antibody HAb-23, which had superior neutralizing activity, for evaluation of its efficacy in SFTSV animal infection models. The results demonstrated that similar to the mouse antibody mAb 40C10, the humanized antibody HAb-23 retained potent therapeutic and protective effects. Although we assessed the degree of humanization and observed a reduction in immunogenicity compared to that of the mouse antibody, the immunogenicity still needs to be further experimentally investigated and clinically evaluated to ensure the safety of the humanized antibody. With the rapid advancement of AI computational methods, the design of humanized antibodies can now begin with the use of predicted structural models and Rosetta atomic simulations to aid in the selection of designs with optimal energy and structural integrity [42], thereby shortening the antibody design cycle. This approach provides us with new technological methods to expedite the humanization of antibodies in the future. However, thorough testing through *in vitro* and *in vivo* experiments is still needed to comprehensively assess the functionality of the designed humanized antibodies.

While the manuscript was under peer-review, Chang et al. reported several neutralizing antibodies targeting SFTSV glycoproteins, including one that interact with domain I of SFTSV

Gn [43]. Structural comparison clearly shows that mAb 40C10 and SF5 recognize different epitopes on domain I, with no overlap between the two (S13A Fig). The affinity and neutralizing activity of mAb 40C10 against Gn surpass those of SF5. This may be because the epitope of SF5 is hidden within the penton/hexon, while the epitope of mAb 40C10 is located on the external surface of the penton/hexon, allowing mAb 40C10 to bind more effectively to each Gn in the penton/hexon (S13B and S13C Fig). In animal experiment, Chang et al. used a challenge dose of 50,000 $LD_{50}$ and tested the protective effects of different antibody doses [43]. A dose of 5 mg/kg of SF5 achieved 100% prophylactic protection, and a dose of 10 mg/kg achieved 100% therapeutic efficacy. Unfortunately, we did not experiment with different doses of mAb 40C10 to determine the minimum dose required for effective prophylactic and therapeutic effects. This aspect will be addressed in future studies to better understand the optimal dosing strategy.

In conclusion, we elucidated the structure of the SFTSV Gn head–mAb 40C10 Fab complex. This study provides insights into the molecular mechanism underlying the broad-spectrum neutralizing activity of mAb 40C10 and reveals a previously unreported potent and broad-spectrum neutralizing epitope of SFTSV Gn, providing a new target for subsequent vaccine and drug design. Additionally, we humanized of mAb 40C10, producing the humanized antibody HAb-23, which maintained potent binding and neutralizing activity. HAb-23 demonstrated effective therapeutic and protective effects in an animal infection model. These findings lay the foundation for the development of antibody therapeutics for clinical applications.

## Materials and methods

### Ethical statement

All animal procedures were performed in strict compliance with the guidelines of Guide for the Care and Use of Laboratory Animals [44]. The animal experiments were approved by the Ethics Committee of Wuhan Institute of Virology, Chinese Academy of Sciences under the approval number WIVA33202006. All cell and animal experiments involving viruses were conducted in Biological safety third-level laboratory (BSL-3) and Animal Biological safety third-level laboratory (ABSL-3).

### Expression and purification of SFTSV Gn head domain

The SFTSV Gn head domain (GenBank accession no. QNR55516.1, residues 20–340) was fused with a C-terminal avi-tag and an eight-histidine purification tag. This construct was subsequently subcloned into the pAcHBM vector, which had been engineered to incorporate an HBM signal sequence at the N-terminus. Transfection and amplification procedures were carried out using Sf9 cells to generate recombinant baculovirus, while the expression of recombinant proteins was achieved in High Five cells.

The cell culture supernatant was harvested 72 h after infection, and the recombinant SFTSV Gn head domain was purified by Ni-NTA affinity chromatography (Smart-Lifescience, SA004100). The affinity column was washed with 100 mL of wash buffer containing 25 mM Tris, 150 mM NaCl, 25 mM imidazole, pH 8.0, and the target protein was eluted with elution buffer (25 mM Tris, 150 mM NaCl, 500 mM imidazole, pH 8.0). The protein was further purified on a Superdex 75 Increase 10/300 GL size-exclusion column (GE Healthcare) equilibrated with 25 mM Tris, 150 mM NaCl, pH 8.0. SDS-PAGE analysis indicated over 98% purity of the final purified recombinant protein. Fractions from the single major peak were pooled and concentrated to 15 mg/mL.

### Preparation of mAb 40C10 Fab fragment

The selected monoclonal cell strain stable producing mAb 40C10 was injected intraperitoneally into mice to produce ascites [20]. The mAb 40C10 IgG from mouse ascites was purified by Protein A affinity chromatography column (GE, Hitrap Protein A HP). The mAb 40C10 IgG was pooled and incubated with papain at a mass ratio of 35:1 at 37 ˚C for 9 h, and further purified on a Protein A affinity chromatography to collect the Fab fragment.

### Analytical gel filtration and purification of the SFTSV Gn head domain–Fab complex

MAb 40C10 Fab was mixed with the SFTSV Gn head domain at a molar ratio of 1:1.2 and incubated on ice for 1.5 h. The complex was further purified on a Superdex 200 Increase 10/300 GL size-exclusion column (GE Healthcare) equilibrated with 25 mM Tris, 150 mM NaCl, pH 8.0, and fractions from the single major peak were pooled and concentrated to 6 mg/mL for crystallization.

### Crystallization

The protein complex was used for crystal screening by vapor-diffusion sitting-drop method, and crystallization screening was performed at 16 ˚C, including PEG/Ion, Crystal Screen, Index, SaltRX from Hampton Research, and Wizard I-IV from Emerald Biosystems. The flake-like crystals appeared after one week in the reservoir solution containing 0.04 M Citric acid, 0.06 M BIS-TRIS propane/pH 6.4, 20% w/v Polyethylene glycol 3,350. The crystals were frozen in liquid nitrogen in reservoir solution supplemented with 25% glycerol (vol/vol) as a cryoprotectant.

### Structure determination and refinement

Diffraction data was collected at the Shanghai Synchrotron Radiation Facility BL10U2 (wavelength, 0.97918 Å) at 100 K. All data sets were processed using DIALS [45]. The structure was determined by molecular replacement using PHASER [46] with the SFTSV Gn structure (PDB ID: 5Y11) [17] and the structures of the Fab fragment available in the PDB with the highest sequence identities. The molecular model was reconstructed with COOT [47]. Subsequently, structural refinement was performed in Phenix [48]. In the final stage of model construction and refinement, glycans and waters were added according to the density map. Data collection and detailed statistics were summarized in S1 Table. Images were made by Pymol (www.pymol.org) and UCSF ChimeraX [49].

### Point mutations at sites of key residues

Based on the three-dimensional structure of the complex, the potential critical binding site of SFTSV Gn was identified, which mediates the interaction between SFTSV Gn and the neutralizing antibody mAb 40C10. Primers were designed to replace the corresponding codons with the GCC sequence, and single amino acid mutation to alanine was introduced using PCR method. The accuracy of point mutation was verified by sequencing, and finally the SFTSV Gn single-point mutant protein was expressed through the Bac-to-Bac baculovirus system. The expression and purification strategy for the SFTSV Gn single-point mutant protein mirrored that of the wild-type SFTSV Gn head domain protein.

### Humanization of antibody

Humanization of antibody was performed using CDR grafting method. The human variable regions that were the most homologous to mouse variable regions as the templates for CDR

grafting of mAb 40C10 were identified and selected through NCBI IgG BLAST. Subsequently, critical residues in the FR of templates that could potentially impact the structure of CDRs were reverted to mutations. Ultimately, the designed heavy and light chains were assessed for their degree of humanization using the "Hu-mAb" website (https://opig.stats.ox.ac.uk/webapps/sabdab-sabpred/sabpred/humab).

Finally, humanized antibodies in the form of human IgG1 were cloned into pCDNA3.1 vector, and a large number of plasmids were extracted. The plasmid was transfected with HEK 293F cells with PEI (Sigma, #408727), incubated at 37 °C, 5% $CO_2$ for 5 days, and the supernatant was collected. Humanized antibodies were purified by Protein A affinity chromatography, and purity was determined by SDS-PAGE.

## Surface plasmon resonance (SPR) assay

The binding kinetics and affinity between mAb 40C10 and SFTSV Gn single-point mutants were measured by SPR (Biacore T200, Cytia). The SPR experiment was conducted in HBS-P buffer (0.01 M Hepes, pH 7.4, 0.15 M NaCl, 0.05% Surfactant P20), using Protein A chip (Cytia, #29127556) to immobilize mAb 40C10 at a flow rate of 10 µl/min. Various concentrations of purified SFTSV Gn single-point mutants, ranging from 400 nM to 1.5625 nM with a 1:2 dilution factor, were then injected at a flow rate of 30 µl/min. The resulting data was analyzed using Biacore Insight Evaluation Software 3.2.1 and fitted to a 1:1 binding model.

The binding kinetics and affinity between SFTSV Gn head domain and humanized antibodies were analyzed by SPR. In SPR analysis, an NTA sensor chip (Cytia, #28994951) was utilized to capture the SFTSV Gn head domain with His tags. The SPR experiment was conducted in HBS-P buffer at a flow rate of 10 µl/min. A series of diluted purified humanized antibodies, with concentrations ranging from 100 nM to 0.78125 nM and a dilution factor of 1:2, were injected. The resulting data were analyzed using Biacore Insight Evaluation Software 3.2.1 and fitted to a 1:1 binding model.

## Neutralization assay

According to previous research [20], first, the antibodies were serially diluted two times in DMEM supplemented with 2% FBS, 100 U/mL penicillin, and 100 µg/mL streptomycin. Subsequently, the diluted antibodies were mixed and incubated with an equal volume of SFTSV at a dose of 100 $TCID_{50}$ at 37 °C for 1 h and added to Vero cells ($10^5$ cells/well) and cultivated in a 96-well plate at 37 °C for 3 days. Finally, SFTSV infection was detected utilizing the polyclonal antibodies (pAb) anti-NPs method, with a control mixture comprising virus and PBS. Hoechst 33258 dye (Beyotime, #C1018) was utilized for nuclear staining. Images were captured, and the quantities of SFTSV-infected cells and total cells per well were determined using the Operetta CLSTM high-throughput system (PerkinElmer). The inhibitory effects post serial dilution were analyzed employing GraphPad Prism software version 8.0.2.

## Animal experiments

In the prophylactic treatment group, 6-week-old female IFNAR$^{-/-}$ C57BL/6 mice (n = 9/group) were administered either 30 µg/g humanized antibody or mouse antibody via intraperitoneal injection. Following the administration of antibodies after 1 day, the mice were intraperitoneally injected with a 50 $LD_{50}$ dose of SFTSV. In the therapeutic treatment group, 6-week-old female IFNAR$^{-/-}$ C57BL/6 mice (n = 9/group) were intraperitoneal injected with SFTSV at a 50 $LD_{50}$ dose of SFTSV. At day 1 post-SFTSV injection, the mice were intraperitoneally administered with 30 µg/g humanized antibody or mouse antibody. An equal volume of PBS was administered as control.

In all groups, body weights and survival rates of mice were monitored and recorded for 12 days after infection. Three mice were sacrificed on the 5th and 12th day after infection, and liver, spleen, lung and kidney were extracted. Tissue sections were prepared, and hematoxylin-eosin (H&E) staining and immunohistochemical assays was performed as previously described [40].

## qRT–PCR

Tissue and serum samples were subjected to lysis and extraction utilizing the MiniBEST Viral RNA/DNA Extraction Kit Ver.5.0 (TaKaRa, #9766) in accordance with the manufacturer's protocol. After RNA extraction, SFTSV RNA load quantification was conducted employing the one-step real-time qRT–PCR and the HiScript II One Step qRT–PCR Probe Kit (Vazyme, #Q222-01) on the Applied Biosystems StepOnePlus (ThermoFisher). The priming and probe sequences were F: 5'-GGGTCCCTGAAGGAGTTGTAAA-3', R: 5'-TGCCTTCACCAAGAC-TATCAATGT-3', HEX-5'-TTCTGTCTTGCTGGCTCCGC-3'-BHQ-2.

## Assessment of mAb 40C10 inhibition effects on the SFTSV infection process

Experiments were conducted using a 24-well plate, with $2 \times 10^5$ Vero cells seeded per well. The cells were infected with SFTSV at a multiplicity of infection (MOI) of 1.

To investigate whether the mAb 40C10 would affect virus attachment to cells, the virus was pre-incubated with either the antibody mAb 40C10 or PBS at 37°C for 1 hour. The mixture was then added to the cells, which were incubated at 4°C for 1 hour. After incubation, the unbound virus was washed away with cold PBS. The cells were collected and TRIzol (Takara) was added to extract RNA.

To further explored whether the mAb 40C10 would affect the virus endocytosis based on the characteristics of SFTSV invasion of cells [39]. The initial stage of virus adsorption followed the same steps as above. After washing away the unbound virus with cold PBS, the cells were cultured in a medium containing 20 mM $NH_4Cl$ at 37°C for one hour. $NH_4Cl$ (20 mM) can inhibit the fusion of the virus with the endosome membrane [40]. The cells were then washed three times with PBS containing 500 μg/mL proteinase K at room temperature to remove the virus that did not enter the cells. The cells were collected and TRIzol (Takara) was added to extract RNA according to the manufacturer's instructions. Then the cDNA was obtained by PrimeScript RT reagent Kit with gDNA Eraser (Perfect Real Time, #RR047A) and the qPCR was detected by TB Green Premix Ex Taq II (Tli RNaseH Plus, #RR820A) from Takara. The relative expression of SFTSV NP (primer as previously shows [20]) and GAPDH (primer: GAPDH-qF: ACCACAGTCCATGCCATCAC, GAPDH-qR: TCCAC-CACCCTGTTGCTGTA) was analyzed. All qRT-PCR tests were performed in triplicates for each sample.

## Supporting information

**S1 Fig. The ratio of the KD value of the SFTSV Gn single-point mutants to the KD value of the SFTSV Gn wild-type (WT).**
(TIF)

**S2 Fig. Sequence alignment among various genotypes of SFTSV and other strains.** The numbers preceding the sequences are the GeneBank IDs of SFTSV strains. QNR55516.1 and AQX34644.1 are the SFTSV strains used for structural and neutralization analyses in this study, both belonging to the C3 genotype. (A) Chinese lineage SFTSV strains. C1 genotype:

HQ141605, HM802203; C2 genotype: KF711946, KF887436, JF906057, JQ317173, KC292316, HQ419232, KR017861; C3 genotype: KC292300, JQ733566, JF951393, KX641916, KX641914; C4 genotype: KF791955, KT254590, HQ419234, HQ141599, KR698340, KR017862; C5 genotype: AB985309, AB985657, JQ670930. (B) Japanese lineage SFTSV strains. J1 genotype: AB985299, AB985300, AB817990, KP663735, AB985298, AB985293; J2 genotype: KR698338, AB985312, AB985295, KP280204, KR698342, KR698343; J3 genotype: AB817989, KR698334, KR698335, KF374684, KJ597824. (C) The sequence alignment of SFTSV with HRTV (Gene-Bank: AIF75092.1) and GTV (GeneBank: ALQ33264.1). The key residues interacting with the heavy chain of mAb 40C10 are highlighted with blue pentagram, and those interacting with the light chain of mAb 40C10 are marked with yellow pentagram. The residue interacting with both the heavy and light chains of mAb 40C10 are marked with black triangle.
(TIF)

**S3 Fig. Overall structure of the SFTSV Gn head–mAb 40C10 complex.** (A and B) The SFTSV Gn head is shown as surface. Domain I of the SFTSV Gn head is colored by sequence variation using the ConSurf server [50,51], while domains II and III are colored gray. The mAb 40C10 heavy and light chains are colored salmon and yellow, respectively. (B) The binding region of mAb 40C10 on the SFTSV Gn head is indicated. The CDR loops of mAb 40C10 colored as described above.
(TIF)

**S4 Fig. Structural comparison between the Gn in the SFTSV Gn head–mAb 40C10 complex crystal structure and the Gn in the cryo-EM structure of the SFTSV virion.** (A and B) The Gn in the SFTSV Gn head–mAb 40C10 complex crystal structure is shown in gray. (A) The Gn-Gc of SFTSV 2-fold hexamer (PDB: 7X6W) is displayed in color. (B) The Gn-Gc of SFTSV 5-fold pentamer (PDB: 7X72) is shown in color.
(TIF)

**S5 Fig. The human germline with high identity to the heavy and light chain of mAb 40C10 sequences retrieved from the database.**
(TIF)

**S6 Fig. Humanness score of the heavy and light chains of human antibody/mouse antibody.** (A) The human antibody (SARS-CoV-2 neutralizing antibody TH132) [52] and the mouse antibody (mAb 40C10) were scored for their degree of humanization using the "HumAb" website. The human antibody serves as a positive control, while the mouse antibody serves as a negative control for humanization scoring.
(TIF)

**S7 Fig. The results of affinity measurement between the humanized antibodies and SFTSV Gn using SPR.**
(TIF)

**S8 Fig. Histopathological analyses of mouse antibody mAb 40C10 treated mice.**
(TIF)

**S9 Fig. Sequence alignment among various genotypes of SFTSV and other strains.** The sequences correspond to those in S2 Fig. The critical binding epitopes of mAb 4–5 are highlighted with black dots, while the binding epitope of Ab10 in the Gn head domain are highlighted with purple triangles.
(TIF)

**S10 Fig. MAb 40C10 neutralizes SFTSV by forming intervirion cross-links.** (A, B and C) The distribution and arrangement of mAb 40C10 on different symmetry axes of SFTSV are shown (EMDB: EMD-35176, EMD-35177, EMD-35178; PDB: 8I4T, 7X6W, 7X72). Black dashed lines indicate the distances between the CH1 domain ends of two adjacent mAb 40C10 Fabs, with other colors consistent with those Fig 2. (D) Neutralization profile of mAb 40C10 IgG or Fab to SFTSV.
(TIF)

**S11 Fig. Binding positions of mAb 4–5 and mAb 40C10 on the penton/hexon of the SFTSV.** Colors are consistent with those in Fig 2. (EMDB: EMD-35176, EMD-35178; PDB: 8I4T, 7X6W, 7X72).
(TIF)

**S12 Fig. MAb 40C10 inhibition effects on the SFTSV infection process.** (A) Binding process and (B) Internalization process.
(TIF)

**S13 Fig. Analysis of binding epitopes of mAb 40C10 and SF5.** (A) The binding epitopes of mAb 40C10 and SF5 on the SFTSV Gn head are shown. SF5 is colored deep pink, other colors consistent with Fig 2. (B and C) MAb 40C10 and SF5 bind to either the hexon or penton Gn protein. Penton, hexon, mAb 40C10 and SF5 are colored accordingly (PDB: 8I4T, 7X6W, and 7X72).
(TIF)

**S1 Table. X-ray diffraction data processing and refinement statistics.**
(DOCX)

**S2 Table. PISA analysis of interaction between SFTSV Gn/mAb 40C10.**
(DOCX)

**S3 Table. Residues contributed to interaction between SFTSV Gn/mAb 40C10.**
(DOCX)

## Acknowledgments

We are grateful to the staffs from BL10U2 beamline at Shanghai Synchrotron Radiation Facility for assistance during data collection.

## Author Contributions

**Conceptualization:** Fei Deng, Shu Shen, Yu Guo, Nan Zhang.

**Funding acquisition:** Xiaoli Wu, Zixian Sun, Fei Deng, Yu Guo, Nan Zhang.

**Investigation:** Pinyi Yang, Xiaoli Wu, Hang Shang, Zixian Sun, Zhiying Wang, Zidan Song, Hong Yuan, Nan Zhang.

**Methodology:** Xiaoli Wu, Shu Shen, Yu Guo, Nan Zhang.

**Project administration:** Fei Deng, Shu Shen, Yu Guo, Nan Zhang.

**Visualization:** Pinyi Yang, Xiaoli Wu, Nan Zhang.

**Writing – original draft:** Pinyi Yang, Xiaoli Wu, Yu Guo, Nan Zhang.

**Writing – review & editing:** Pinyi Yang, Xiaoli Wu, Hang Shang, Zixian Sun, Zhiying Wang, Zidan Song, Hong Yuan, Fei Deng, Shu Shen, Yu Guo, Nan Zhang.

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
