## [Decision Letter · Decision Letter 0]

15 Jul 2024

Dear Dr. Zhang,

Thank you very much for submitting your manuscript "Molecular mechanism and structure-guided humanization of a broadly neutralizing antibody against SFTSV" for consideration at PLOS Pathogens. As with all papers reviewed by the journal, your manuscript was reviewed by members of the editorial board and by several independent reviewers. The reviewers appreciated the attention to an important topic. Based on the reviews, we are likely to accept this manuscript for publication, providing that you modify the manuscript according to the review recommendations.

Three experts in the field have provided feedback and have several comments and questions that should be addressed prior to further consideration of the manuscript. Among these are the mechanism of neutralization and considerations of the icosahedral nature of the SFTSV vision, and the ability of antibodies to bind across subunits. There are also some grammar and typos that should be corrected.

Sincerely,

Richard J. Kuhn, PhD

Academic Editor

PLOS Pathogens

Kanta Subbarao

Section Editor

PLOS Pathogens

Michael Malim

Editor-in-Chief

PLOS Pathogens

orcid.org/0000-0002-7699-2064

Three experts in the field have provided feedback and have several comments and questions that should be addressed prior to further consideration of the manuscript. Among these are the mechanism of neutralization and considerations of the icosahedral nature of the SFTSV vision, and the ability of antibodies to bind across subunits. There are also some grammar and typos that should be corrected.

Reviewer Comments (if any, and for reference):

Reviewer's Responses to Questions

**Part I - Summary**

Reviewer #1: In this manuscript, Yang et al. employed X-ray crystallography to determine the crystal structure of the previously studied monoclonal antibody 40C10 in complex with the SFTSV Gn antigen. They analyzed the neutralizing epitope and, based on structural information, performed humanization of the antibody. The affinity and neutralizing activity of the humanized antibodies were subsequently evaluated, and the humanized antibodies retained significant activity. Additionally, the preventive and therapeutic effects of the humanized antibody were evaluated in an animal infection model.

Overall, this manuscript reveals a novel binding epitope through the structural analysis of the complex of antibody 40C10 and SFTSV Gn, elucidating the molecular mechanism of its broad-spectrum neutralizing activity. The epitope was found to be conserved across different genotypes of SFTSV and SFTSV-related viruses. To enhance clinical applicability, the authors humanized of antibody 40C10, and the results showed that the humanized antibodies retained potent neutralizing activity and provided effective protection in an animal infection model, laying a foundation for clinical application. The study is clear and offers new directions for future SFTSV antibody therapies, vaccines, and drug design. The context fits well into the scope of this journal. However, some issues in the manuscript need minor revisions and discussion to enhance the precision of the content.

1. In this manuscript, regarding the structural analysis of the antibody 40C10 and SFTSV Gn, the authors mentioned in line 193 that R106 of HCDR3 forms a hydrogen bond with the glycan molecule of the N-linked glycosylation site N63 of Gn. Is the glycosylation site N63 of Gn conserved across different genotypes of SFTSV? Does this interaction also occur in different genotypes of SFTSV? Interactions between antibodies and glycan molecules of antigens are relatively rare. Therefore, the authors should further analyze in the discussion section whether this interaction is conserved or specific to this particular antibody-antigen pair.

2. The authors previously found that the antibody 40C10 exhibited cross-neutralizing activity against HRTV and GTV and analyzed the potential molecular mechanisms underlying this cross-reactivity through structural and sequence analyses. However, they did not experimentally verify whether the humanized antibody HAb-23 retains this cross-neutralizing activity. Conducting experiments to validate that the humanized antibody still possesses cross-neutralizing activity would strengthen the conclusions of the study.

3. In Fig 3D, the authors used log2 for the x-axis to represent antibody concentration. However, antibody concentration should typically be expressed in log10 rather than log2, as this facilitates easier interpretation of the results.

4. In the animal experiment described in line 302, the authors used a dose of 50 LD50 for infecting mice. Why was the infection dose chosen as LD50 instead of using TCID50 or PFU?

5. Recently, a paper titled "Bispecific antibodies targeting two glycoproteins on SFTSV exhibit synergistic neutralization and protection in a mouse model" was published in PNAS. In this study, several antibodies were isolated from an SFTS convalescent patient, and structural analysis revealed that one of these antibodies, SF5, binds to subdomain I of the SFTSV Gn. However, upon comparison, it was found that the specific epitope of SF5 differs from the antibody 40C10 reported in this manuscript. It is recommended to update the discussion section with this latest research. A detailed comparison and discussion of both the structural and animal experiment results should be included.

6. There are also formatting issues in the manuscript that need attention: For instance, in line 205, the sentence “… exhibited a 3-5-fold decrease…” should use an en dash (–) to represent "to", thus it should be written as "3–5." Similarly, in lines 88-89 and line 201, the sentence "C1-C5" and "J1-J3" should also use an en dash, becoming "C1–C5" and "J1–J3," respectively.

Additionally, in line 306, the legend of Fig 4 needs to correctly format the mouse type symbols.

Reviewer #2: The manuscript titled “Molecular mechanism 1 and structure-guided humanization of broadly neutralizing antibody against SFTSV” by Pinyi Yang and colleagues performed structural studies on a previously reported monoclonal antibody called 40C10 against the Severe fever with thrombocytopenia syndrome virus. The crystal structure of antibody 40C10 in complex with SFTSV glycoprotein N revealed a novel antibody targeting site on Gn that has not been reported previously. This epitope, located on domain I of Gn head, is highly conserved among subtypes, explaining the broad neutralization by 40C10. The authors then performed humanization on this antibody and obtained 3 heavy and 3 light chains, respectively. They expressed 9 antibodies by combinations of new heavy and light chains and performed binding and neutralizations and identified one antibody to test its prophylactic and therapeutic potentials. Overall, the authors identified a new antibody targeting site on Gn and obtained humanized version of this mouse antibody.

Reviewer #3: The authors describe the structural basis of epitope recognition of a potently neutralizing murine antibody 40C10 on SFTSV. The epitope is on the Gn glycoprotein domain I of its head region. The authors also provide evidence of the potential therapeutic applicability of the humanized version of the mAb. Taken together, this work adds to the growing list of anti-viral antibodies currently under development against many different viruses and is of broad interest to the audience of PLOS Pathogens.

**Part II – Major Issues: Key Experiments Required for Acceptance**

Reviewer #1: see the report.

Reviewer #2: None

Reviewer #3: The authors describe the structural basis of epitope recognition of the murine antibody 40C10 on the head portion of the viral glycoprotein Gn using X-ray crystallography. Additionally, they produce humanized version of this mAb which maintain potent neutralization capability in vitro as well as in vivo.

However, there are a a few aspects that if addressed can enrich this manuscript significantly.

1. The authors fit the crystal structures of the Fab + Gn head domain onto the existing model of the STSFV determined by Cryo-EM. They show that the Fab does not have any major clashes at either on the hexon or penton positions and therefore, in principle can saturate all the binding sites. Could the authors suggest whether the mAb 40C10 can form intra-/inter-virion crosslinks on the viral surface based on the distance criteria between the Fab arms of an IgG? The authors should check the following paper from 2023: DOI: 10.1073/pnas.2213690120. The paper delves into the possible link between neutralization potencies of neutralizing mAbs and their arrangement on icosahedral viruses. The authors should also do the same analysis with the other mAb (mAb 4-5) for which a structural model exists. It would be interesting to see if both mAbs 40C10 and 4-5 can form intra-virion crosslink or not. The reason I am suggesting this is because SFTSV is icosahedral.

2. The epitope recognized by 40C10 is conserved among different strains of SFTSV as well as among HRTV and GTV. The Multiple sequence alignment in Fig S2 shows significant similarity among the different strains. Could the authors also show the residues that constitute the footprint for mAb 4-5? If so, then it should also exhibit broad neutralizing capability against different strains. The authors should discuss at least a bit about the extent of sequence conservation of the three different epitopes targeted by the three antibodies, mAb 40C10, mAb 4-5 and Ab10.

3. What is the mechanism of action of this antibody? Does it block attachment or receptor engagement or fusion? there is some discussion about this, but no experimental evidence. It would really enrich the manuscript if the authors can show if the antibody blocks either attachment or receptor engagement or fusion or all of them. If it is currently beyond the scope of the study, then if they could show that the antibody is able to block before or after attachment to cells, then that would be helpful too.

If these points are addressed, then I am happy to recommend the paper for publication in PLOS Pathogens.

**Part III – Minor Issues: Editorial and Data Presentation Modifications**

Reviewer #1: see the report.

Reviewer #2: 1. Docking the 40C10 structure on virion structure indicated the Fab binding is compatible with both penton and hexon. can the authors show RMSD on alignment of Gn in crystal structure and in virion? Can the authors discuss what is the potential mechanism of 40C10 against SFTSV infection?

2. Is it possible to show in Figure 1 as another panel in which the Gn is shown in surface representation and colored by sequence variation with the footprint of 40C10 highlighted.marked?

2. The authors claimed they have "humanized" this antibody based on a website, it will be helpful to show the humanness score of the mouse version as a "negative control" and may use another know human antibody as a "positive control"

3. The author did not provide any experimental data on the "humanness", for example, the pharmacokinetics comparison in a model system, such as the Human FcRn transgenic mice.

4. Line 233 typo "exit" should be "exist"

Reviewer #3: There are some grammatical mistakes. Please do a word check.

PLOS authors have the option to publish the peer review history of their article (what does this mean?). If published, this will include your full peer review and any attached files.

Reviewer #1: **Yes: **wei liu

Reviewer #2: No

Reviewer #3: No

Figure Files:

Data Requirements:

Reproducibility:

References:

---

## [Editor Report · Decision Letter 1]

1 Sep 2024

Dear Dr. Zhang,

We are pleased to inform you that your manuscript 'Molecular mechanism and structure-guided humanization of a broadly neutralizing antibody against SFTSV' has been provisionally accepted for publication in PLOS Pathogens.

Best regards,

Richard J. Kuhn, PhD

Academic Editor

PLOS Pathogens

Kanta Subbarao

Section Editor

PLOS Pathogens

Michael Malim

Editor-in-Chief

PLOS Pathogens

orcid.org/0000-0002-7699-2064

The revised version addresses all of the comments and queries from the three reviewers and they have substantially improved the manuscript.
---

## [Editor Report · Acceptance letter]

12 Sep 2024

Dear Dr. Zhang,

We are delighted to inform you that your manuscript, "Molecular mechanism and structure-guided humanization of a broadly neutralizing antibody against SFTSV," has been formally accepted for publication in PLOS Pathogens.

Best regards,

Michael Malim

Editor-in-Chief

PLOS Pathogens

orcid.org/0000-0002-7699-2064